# Understanding current practice, identifying barriers and exploring priorities for adverse event analysis in randomised controlled trials: an online, cross-sectional survey of statisticians from academia and industry

Rachel Phillips 🔘 , Victoria Cornelius

Faculty of Medicine, School of Public Health, Imperial College London, London, UK

**Correspondence to**
Rachel Phillips;
r.phillips@imperial.ac.uk

## ABSTRACT

**Objectives** To gain a better understanding of current adverse event (AE) analysis practices and the reasons for the lack of use of sophisticated statistical methods for AE data analysis in randomised controlled trials (RCTs), with the aim of identifying priorities and solutions to improve practice.

**Design** A cross-sectional, online survey of statisticians working in clinical trials, followed up with a workshop of senior statisticians working across the UK.

**Participants** We aimed to recruit into the survey a minimum of one statistician from each of the 51 UK Clinical Research Collaboration registered clinical trial units (CTUs) and industry statisticians from both pharmaceuticals and clinical research organisations.

**Outcomes** To gain a better understanding of current AE analysis practices, measure awareness of specialist methods for AE analysis and explore priorities, concerns and barriers when analysing AEs.

**Results** Thirty-eight (38/51; 75%) CTUs, 5 (5/7; 71%) industry and 21 attendees at the 2019 Promoting Statistical Insights Conference participated in the survey. Of the 64 participants that took part, 46 participants were classified as public sector participants and 18 as industry participants. Participants indicated that they predominantly (80%) rely on subjective comparisons when comparing AEs between treatment groups. Thirty-eight per cent were aware of specialist methods for AE analysis, but only 13% had undertaken such analyses. All participants believed guidance on appropriate AE analysis and 97% thought training specifically for AE analysis is needed. These were both endorsed as solutions by workshop participants.

**Conclusions** This research supports our earlier work that identified suboptimal AE analysis practices in RCTs and confirms the underuse of more sophisticated AE analysis approaches. Improvements are needed, and further research in this area is required to identify appropriate statistical methods. This research provides a unanimous call for the development of guidance, as well as training on suitable methods for AE analysis to support change.

### Strengths and limitations of this study

► A high response rate was achieved from UK Clinical Research Collaboration clinical trial unit and industry statisticians invited to participate in this survey.
► There was some level of self-selection to participation and as such, there is a possibility that participants had an increased interest in adverse event analysis and are not fully representative of the clinical trial community.
► The survey was followed up with a workshop of senior statisticians from across the UK, which represents more of a general interest group.
► The survey provides insight and essential starting points to identify areas of focus to help support a change to improve adverse event analysis practices.

## INTRODUCTION

Randomised controlled trials (RCTs) are a valuable source of information when establishing the harm profile of medicinal products. They provide a controlled comparison of adverse event (AE) rates, thus allowing causality to be evaluated and potential detection of adverse drug reactions. Adverse events are events that may or may not be related to the treatment under investigation, and adverse drug reactions are events classified as related to the treatment under investigation. (An adverse event is defined as 'any untoward medical occurrence that may present during treatment with a pharmaceutical product but which does not necessarily have a causal relationship with this treatment'. An adverse drug reaction is defined as 'a response to a drug which is noxious and unintended …' where a causal relationship is 'at least a reasonable possibility'.) Reviews of published RCT reports have demonstrated that harms

data are not being analysed to its full potential.[1–5] Most notable inadequacies include ignoring information on repeated events and dichotomising continuous clinical and laboratory outcomes, with binary counts often presented using simple tabulations, indicating whether an event did or did not occur. Little formal analysis is performed but a comprehensive methods review undertaken by the authors revealed that there have been many published statistical methods proposed specifically to analyse adverse event data for both the interim and the final analysis. These include using time-to-event approaches, Bayesian methods that can incorporate prior information and visual analysis.[6 7] Many of the proposed methods could be adopted into current practice with relative ease. Chuang-Stein and Xia[8] have proposed examples of industry strategies adopting such methods. Previous research has demonstrated that these methods are not used for the analysis presented in the primary results publication. In a recent systematic review of 184 published reports in high impact general medical journals, there are no examples of these proposed methods being used, with authors preferring simple approaches predominantly presenting frequencies and percentages of events.[1 5] The statistical methods proposed for adverse event analysis identified in the methodology review also had minimal citations, which further suggests that uptake of these methods is low.[1 6 7]

In addition, there is a problem with the reporting of adverse events and the selection of events to include in journal articles. Many reviews have established poor quality reporting in journal articles of adverse event data from RCTs.[9–15] Also it is often not possible to include all adverse events in the primary RCT publication and authors need to select events for a pertinent summary. To achieve this, there is a prevalent practice of relying on arbitrary rules to select events to report, which can introduce reporting biases leaving out important adverse events. This also creates a barrier to establishing an accurate harm profile.[3 16]

Understanding the reasons for the low uptake of these statistical methods will help identify solutions to improve the analysis of adverse events in RCTs. We undertook a survey of UK statisticians working in clinical trials to investigate their current practice when analysing adverse events, to measure their awareness of available methods for adverse event analysis and to explore their priorities, concerns and identify any perceived barriers when analysing adverse events.

## METHODS
### Study design
A cross-sectional, online survey of UK Clinical Research Collaboration (CRC) clinical trial unit (CTU) and industry statisticians from both pharmaceuticals and clinical research organisations (CROs) was conducted. We aimed to recruit a minimum of one statistician from each of the 51 UKCRC registered CTUs and from a sample of

pharmaceutical companies and CROs in the UK to gain an industry perspective. The survey was followed up with a workshop at the UKCRC biannual statisticians' operations group meeting where survey results were presented and areas for improvements and priorities were discussed.

### Survey development
The survey was developed using information from current guidance and previous research that examined barriers to the uptake of new methodology.[17–20] Topics covered included questions about current practice and factors influencing adverse event analysis performed, barriers encountered when analysing adverse events, concerns regarding adverse event analysis, awareness and opinions of specialist methods for adverse event analysis, concerns and barriers of implementing specialist methods, and opinions on potential solutions to support a change in adverse event analysis practice.

Questions were predominantly closed form but where appropriate open-ended questions were included to allow for detailed responses and comments. Responses were measured using Likert scales. Survey questions for UKCRC CTU and industry statisticians were identical (online supplementary appendix item 1). The survey was piloted on clinical trial statisticians (n=6) at three CTUs prior to launching nationwide to ensure understanding of the questions, whether sufficient response categories had been included, and if certain questions were consistently left unanswered, as well as the usability and functionality of the online platform hosted by SurveyMonkey.[21]

### Sampling and recruitment
We targeted a population that we knew to be predominantly involved in the analysis of adverse events in clinical trials. Specifically, the UKCRC CTU Statistics Operation network supported the survey and contacted each of the 51 registered CTUs' senior statisticians on behalf of the study team. Email invitations were also sent directly to a convenience sample of seven senior statistical contacts working in UK-based pharmaceuticals (AstraZeneca, Boehringer Ingelheim, Glaxo-Smith-Kline (GSK), Novartis and Roche) and CROs (Cytel and IQVIA). The invitations requested that one statistician within the unit or organisation complete the survey. Reminder emails were sent to non-responders. The survey opened in April 2019 and remained open for 8 weeks. We also created an open platform for participants that was promoted at the June 2019 Promoting Statistical Insights (PSI) conference, the Effective Statistician podcast broadcast in July 2019, and Twitter and LinkedIn platforms. This platform remained open for 10 weeks. Participants that successfully completed the survey were automatically entered into a prize draw to win £50 worth of gift vouchers.

The invitation to participate in the study included the participant information sheet (online supplementary appendix item 2), which was also included at the beginning of the survey before participants formally entered. Participants were encouraged to read the information

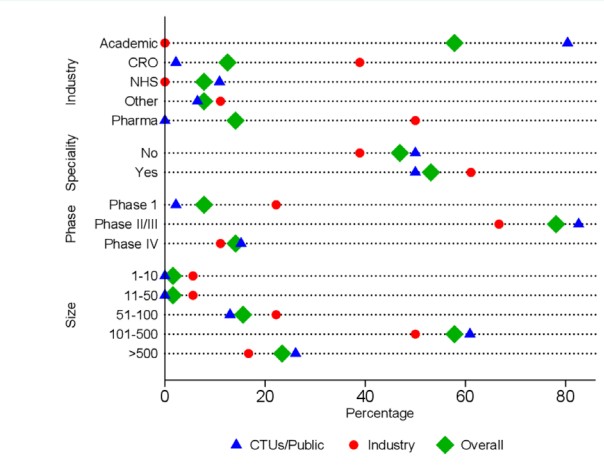

**Figure 1** Participant characteristics by sector and overall. CRO, clinical research organisation; CTUs, clinical trials units; pharma, pharmaceuticals.

sheet and discuss the study with others or contact the research team if they wished. If invitees were happy to enter into the trial at that point, their consent was taken as implied on submission of the completed survey.

### Participants
Statisticians with experience of planning and preparing the final analysis reports for pharmacological RCTs were invited to participate.

### Analysis
Descriptive analysis was undertaken, primarily including frequencies and proportions for each questionnaire item and where appropriate was accompanied with visual summaries.[22] The frequency and proportion of participants that showed support for an item were calculated by combining the 'always' and 'often' or 'strongly agree' and 'agree' categories. Participants were classified according to affiliation into either CTU/public sector or industry sector and analysis was stratified by sector. Response rates were calculated for groups of participants where known.

### Patient and public involvement
This survey forms part of a wider research project that was developed with input from a range of patient representatives. There were no patients directly involved in this survey, but the original proposal and patient and public involvement (PPI) strategy were reviewed by service user representatives (with experience as clinical trial participants and PPI advisors) who provided advice specifically with regard to communication and dissemination to patient and public groups.

### RESULTS
#### Participant flow
Invitations were sent to 51 CTU/public sector and 7 industry contacts. Thirty-eight (75%) units and 5 (71%) industry contacts participated in the survey, giving

an overall response rate of 74%. Twenty-four people consented to participate via the open platform, of which 21 participated in the survey. Eight of which were included in the CTU/public sector group and 13 in the industry sector. In total, 64 participants took part in the survey with n=46 from the CTU/public sector and n=18 from industry (online supplementary appendix figure A1).

#### Participant characteristics
Overall, more than 80% of responders worked on studies of more than 100 participants and nearly 80% worked on phase II/III trials. A greater proportion of industry participants were working on phase I/dose-finding trials compared with CTU/public sector participants (22% vs 2%) (figure 1). The mean number of years of experience was 12.8 (SD 8.3) (median 11.5 years, range (1–35 years)) (table 1).

#### Current analysis practice
Seventy-five per cent of participants reported that they present both 'the number of participants with at least one event' and 'the number of events', 13% reported only presenting 'the number with at least one event', 2% stated that they only present 'the number of events' and 11% reported not presenting either of these (table 2 and online supplementary appendix table A1 for free text comments).

Ninety per cent of participants reported that they use frequencies and percentages to summarise adverse event data, less than 20% reported use of risk differences (16%), odds ratios (16%) or risk ratios (17%), just under a quarter reported use of incidence rate ratios (23%) (table 2). Several participants included comments (n=5) that the summary statistic used for analysis depended on the specific study being analysed.

When comparing adverse event rates between treatment arms, 80% of participants reported typically relying on subjective comparisons, 33% compare rates using hypothesis tests and 22% use 95% CIs as a means to examine the null hypothesis of no difference. CTU/public sector participants reported wider use of both hypothesis tests (39% CTUs/public sector vs 17% industry) and 95% CIs (26% CTUs/public sector vs 11% industry) (table 2). Four free text comments cautioned against the use of testing.

Just under 40% stated that they were aware of appropriate methods published specifically for adverse event analysis in RCTs (table 2). There were five broad groups of methods mentioned, including Bayesian methods to analyse low frequencies (n=1); standard regression modelling approaches such as Poisson, negative binomial and survival approaches (n=6); methods to analyse incidence rates (n=5); meta-analysis approaches for rare events (n=2); and graphical approaches (n=2) (full text comments in online supplementary appendix table A2). Participants also directed us to theoretical and applied examples in the literature (n=6) (full free text comments in online supplementary appendix table A2).[18 23–27]

**Table 1** Participant characteristics by sector and overall

| Characteristics | | CTU/public (N=46) | | Industry (N=18) | | Overall (N=64) | |
|---|---|---|---|---|---|---|---|
| | | n/N | % | n/N | % | n/N | % |
| Typical trial size | 1–10 | 0/46 | 0.0 | 1/18 | 5.6 | 1/64 | 1.6 |
| | 11–50 | 0/46 | 0.0 | 1/18 | 5.6 | 1/64 | 1.6 |
| | 51–100 | 6/46 | 13.0 | 4/18 | 22.2 | 10/64 | 15.6 |
| | 101–500 | 28/46 | 60.9 | 9/18 | 50.0 | 37/64 | 57.8 |
| | >500 | 12/46 | 26.1 | 3/18 | 16.7 | 15/64 | 23.4 |
| Work setting | Academic institution | 38/46 | 82.6 | 0/18 | 0.0 | 38/64 | 59.4 |
| | CRO | 1/46 | 2.2 | 7/18 | 38.9 | 8/64 | 12.5 |
| | NHS trust | 5/46 | 10.9 | 0/18 | 0.0 | 5/64 | 7.8 |
| | Pharmaceutical | 0/46 | 0.0 | 9/18 | 50.0 | 9/64 | 14.1 |
| | Other | 2/46 | 4.3 | 2/18 | 11.1 | 4/64 | 6.3 |
| Speciality* | No | 23/46 | 50.0 | 7/18 | 38.9 | 30/64 | 46.9 |
| | Yes | 23/46 | 50.0 | 11/18 | 61.1 | 34/64 | 53.1 |
| Typical trial phase | Phase I/dose finding | 1/46 | 2.2 | 4/18 | 22.2 | 5/64 | 7.8 |
| | Phase II/III | 38/46 | 82.6 | 12/18 | 66.7 | 50/64 | 78.1 |
| | Phase IV | 7/46 | 15.2 | 2/18 | 11.1 | 9/64 | 14.1 |
| Years of experience | Mean (SD) | 12.0 | (7.2) | 14.7 | (10.7) | 12.8 | (8.3) |
| | Median (min, max) | 12.0 | (1, 30) | 15.5 | (1, 35) | 11.5 | (1, 35) |

*Participants were asked if there was a clinical area they predominantly worked on.
CRO, clinical research organisation; CTU, clinical trial unit; max, maximum; min, minimum; SD, standard deviation.

Only 13% reported undertaking specialist adverse event analysis (table 2), of which five participants provided details. Two reported use of time-to-event approaches, one used data visualisations, one used Bayesian methods and one incorporated repeated events (full free text comments are reported in online supplementary appendix table A3).

Of the participants who reported that they were aware of specialist adverse event analysis methods, we asked opinions on why such methods were not more widely used. Just over a quarter thought limited use was due to technical complexity (27%); over a third thought it could be due to trial characteristics such as unsuitability of sample sizes (36%) and the number of different adverse events experienced in trials (36%); and 46% thought methods were too resource-intensive and methods were not suitable for typical adverse event rates observed (online supplementary appendix table A4).

Over three-quarters (77%) of participants provided further reasons for lack of use of specialist methods. Reasons were characterised into comments relating to: concerns with the suitability of methods in relation to trial characteristics and nature of adverse event data (n=7); opposition and a lack of understanding from clinicians (n=5); a lack of need for such methods (n=3); a desire to keep analysis consistent with historical analysis (n=3); and training and resources (n=1) (online supplementary appendix table A5).

### Influences, barriers and opinions

The most common influences for the adverse event analysis performed were cited as the chief investigator's preference for simple approaches (78%), the observed adverse event rates (76%) and the size of the trial (73%). Over 60% of participants felt that the statistician preferred simple approaches for adverse event analysis (68%), and the number of different adverse events experienced in a trial was influential (65%). Less than 50% of participants thought that journals (48%) or regulators (48%) preferred simple approaches, but there was a notable difference by sector. A greater proportion of industry participants thought regulators preferred simple approaches (67% vs 40%); and a greater proportion of CTU/public sector participants thought journals preferred simple approaches (56% vs 28%) (figure 2 and online supplementary appendix table A6).

Seventy-nine per cent of participants indicated that there are a lack of training opportunities to learn what methods are appropriate for adverse event analysis, two-thirds (66%) believed that there is a lack of awareness of appropriate methods and 58% believed that there is a lack of knowledge to implement appropriate methods. Approximately 60% of participants thought that trial characteristics including trial sample size (61%), number of different adverse events experienced (61%) and adverse event rates (65%) were barriers when analysing such data. Only a third (34%) of participants agreed that

**Table 2** AE information typically presented by sector and overall

| | CTU/public (N=46) | | Industry (N=18) | | Overall (N=64) | |
|---|---|---|---|---|---|---|
| | n/N | % | n/N | % | n/N | % |
| **Information presented** | | | | | | |
| Number of participants with at least one event | 4/46 | 8.7 | 4/18 | 22.2 | 8/64 | 12.5 |
| Number of events | 1/46 | 2.1 | 0/18 | 0.0 | 1/64 | 1.6 |
| Both of the above | 36/46 | 78.3 | 12/18 | 66.7 | 48/64 | 75.0 |
| None of the above | 5/46 | 10.9 | 2/18 | 11.1 | 7/64 | 10.9 |
| Other* | 16/46 | 34.8 | 6/18 | 33.3 | 22/64 | 34.4 |
| **Descriptive and summary statistics†** | | | | | | |
| Frequencies | 42/46 | 91.3 | 16/18 | 88.9 | 58/64 | 90.6 |
| Percentages | 43/46 | 93.5 | 14/18 | 77.8 | 57/64 | 89.1 |
| Risk difference | 5/46 | 10.9 | 5/18 | 27.8 | 10/64 | 15.6 |
| Odds ratio | 7/46 | 15.2 | 3/18 | 16.7 | 10/64 | 15.6 |
| Risk ratio | 6/46 | 13.0 | 5/18 | 27.8 | 11/64 | 17.2 |
| Incidence rate ratio‡ | 8/46 | 17.4 | 7/18 | 38.9 | 15/64 | 23.4 |
| Other§ | 6/46 | 13.0 | 4/18 | 22.2 | 10/64 | 15.6 |
| **AE comparison†** | | | | | | |
| Subjective comparison | 36/46 | 78.3 | 15/18 | 83.3 | 51/64 | 79.7 |
| Exclusion of null through 95% CI | 12/46 | 26.1 | 2/18 | 11.1 | 14/64 | 21.9 |
| Hypothesis test/p value | 18/46 | 39.1 | 3/18 | 16.7 | 21/64 | 32.8 |
| Other¶ | 4/46 | 8.7 | 5/18 | 27.8 | 9/64 | 14.1 |
| **Awareness of any published methods specifically to analyse AEs** | | | | | | |
| No | 25/44 | 56.8 | 4/17 | 23.5 | 29/61 | 47.5 |
| Yes | 11/44 | 25.0 | 12/17 | 70.6 | 23/61 | 37.7 |
| Don't know | 8/44 | 18.2 | 1/17 | 5.9 | 9/61 | 14.8 |
| **Undertaken any specialist AE analysis not mentioned in your previous response** | | | | | | |
| No | 38/43 | 88.4 | 14/17 | 82.4 | 52/60 | 86.7 |
| Yes | 5/43 | 11.6 | 3/17 | 17.6 | 8/60 | 13.3 |

*Other ways of presenting AE information included presenting information on: overall number of events (n=2); number of patients experiencing 0, 1, 2, etc, events and number of AEs per patient (n=2); duration (n=1); relatedness (n=1) and severity (n=7) (full free text comments in online supplementary appendix table A1).
†Participants were able to provide multiple responses to this question.
‡Incorporates free text comments that described summaries synonymous with incidence rate ratios.
§Included a comment that a participant presents the 'median number (IQR)' of events.
¶Other comments related to the calculation of CIs for precision (n=2), one indicated use of a graphical summary (n=1) and four cautioned against the use of testing.
AE, adverse event; CI, confidence interval; CTU, clinical trial unit.

a lack of statistical software/code to implement appropriate methods was a barrier (figure 2 and online supplementary appendix table A7).

The majority of participants (84%) held the opinion that there are a lack of examples for appropriate analysis methods in the applied literature and 44% of participants thought that there are a lack of appropriate analysis methods. Over half of participants indicated that statisticians (69%), journals (60%) and chief investigators (52%) do not give adverse event data the same priority as the primary efficacy outcome. Only 13% of participants believe that regulators do not prioritise adverse event data, but nearly a quarter (24%) felt unable to comment on regulators' priorities (figure 2 and online supplementary appendix table A8).

### Concerns and solutions

When participants were asked to think about available methods for adverse event analysis, the most common concern, which was held by 38% of participants, was acceptability of methods to regulators. This differed substantially by sector with only 23% of CTU/public sector participants holding this belief compared with 77% of industry participants. Twenty per cent of participants were concerned about the acceptability of methods to the chief investigator and journals and 32% were concerned

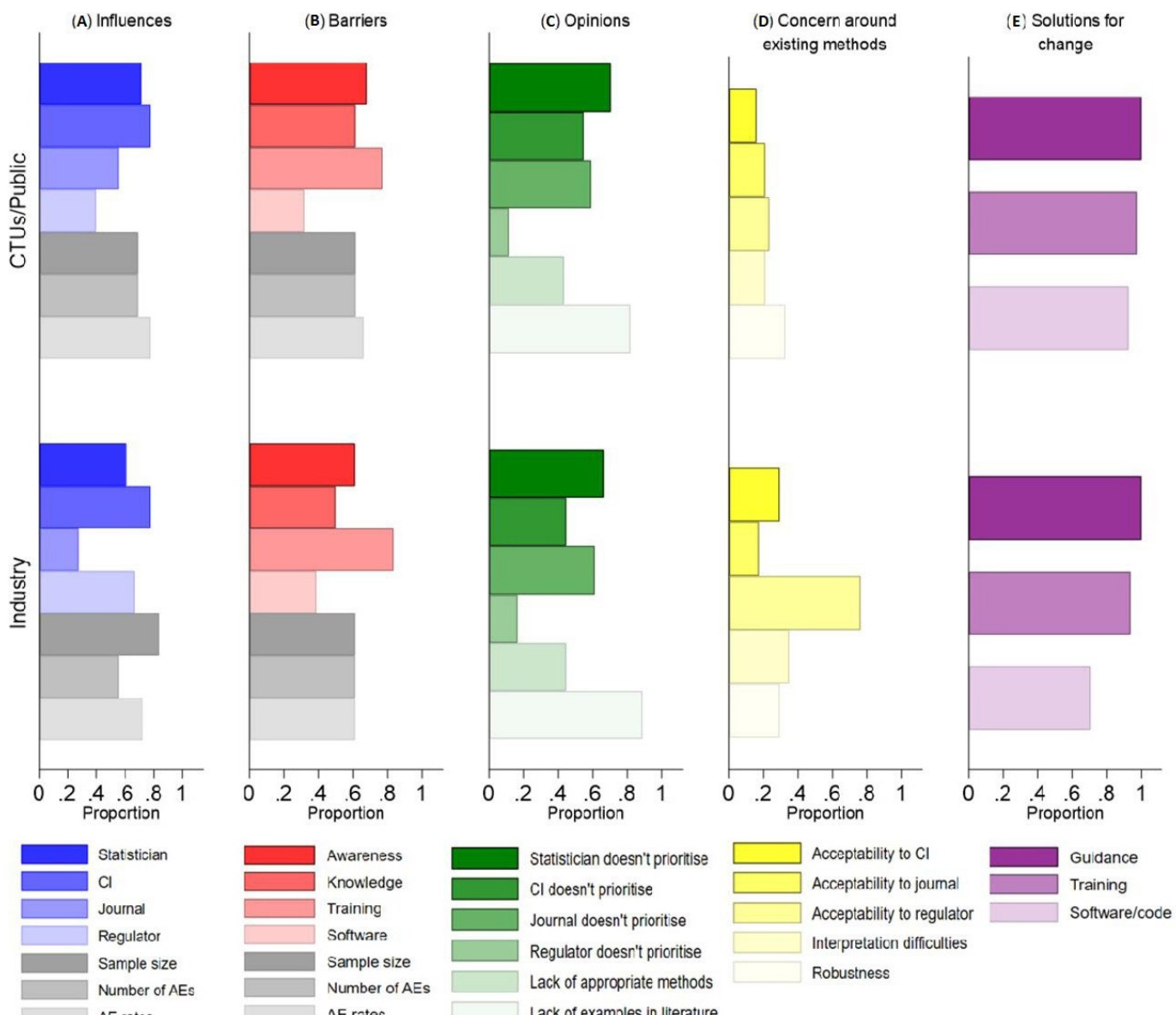

**Figure 2** Survey results by sector: (A) influences on the analysis of AEs, (B) barriers to improve AE analysis, (C) opinions on current AE analysis, (D) reasons for concern with existing methods for AE analysis and (E) potential solutions for change (improving AE analysis). AE, adverse event; CI, chief investigator; CTU, clinical trials unit.

about the robustness of methods (figure 2 and online supplementary appendix table A9).

All participants believed that guidance on appropriate adverse event analysis is needed, 97% thought training specifically for adverse event analysis is needed and 63% thought new software or code is needed (figure 2 and online supplementary appendix table A10). Just under a third (32%) of participants offered solutions to support change in adverse event analysis practices. These included suggestions regarding improved standards or calls for changes from journals, registries and regulators (n=8); development of guidance, education and engaging with the medical community (n=9); and analysis (n=3) (online supplementary appendix table A11).

Thirty per cent of participants raised other items not listed in the survey regarding current adverse event

analysis practices; these covered the following themes: minimum summary information that participants would expect to be reported for adverse event data such as 'numbers and percentages' (n=2); changes to analysis practice that could or have been made such as 'use of graphical methods' (n=8); concerns about the quality and collection of adverse event data (n=3); and general comments and criticisms about current adverse event analysis and reporting practices (n=4) (online supplementary appendix table A12).

In the follow-up workshop of senior statisticians (n=52 from 43 UKCRC registered CTUs) attending the UKCRC biannual statisticians' operations meeting in November 2019, participants were asked to rate the need to improve analysis practices for adverse event data on a scale of 0–100 (indicating low to high priority). The mean score was 66

(SD 16.2) (median 71 (range 9–88)) (n=44). In discussions, the following themes were highlighted as priorities to take forward: development of guidelines; identification of appropriate analysis methods; exploring integration of qualitative information; and ensuring consistency of information reported including development of core harm outcomes by drug class.

## DISCUSSION

Despite RCTs being a valuable source of data to compare rates of adverse events between treatment groups and provide an opportunity to assess causality, analysis and reporting practices are often inadequate.[1–4 9–15] This survey of statisticians from the UK public and private sectors has established a more detailed picture of clinical trial statisticians' adverse event analysis practices and builds on our previous research that evaluated adverse event analysis practices reported in journal articles.[1] It has identified priorities and concerns including influences, barriers and opinions to be addressed in future work to improve adverse event analysis.

Results were broadly similar across public and industry sectors with the only notable differences being the greater use of hypothesis testing and 95% CIs as a means to compare adverse event rates between treatment groups by CTU participants, a more predominant belief by industry participants that regulators preferred simple approaches to adverse event analysis, and a greater concern about acceptability of methods to regulators by industry participants. Across sectors, there was unanimous support that guidance and training on appropriate adverse event analysis are needed.

Survey responses indicated that 75% of statisticians produce tables with both the number of participants with at least one event and the total number of events. This is substantially higher than that reported in reviews of published articles, which found between 1% and 9% reported both.[1–3] The number of total events experienced can give a better summary of impact to patients' quality of life, but it seems this is often omitted from journal articles with reviews identifying only 6% to 7% of published articles reporting this information.[1 4] Reported use of 95% CIs was similar to that reported in journal articles (22% compared with 20%) but reported use of hypothesis testing was lower than what was found in journal articles (32% compared with a range of 38%–47%).[1–3] Reasons for these disparities are not known but could include journal editors requesting such analyses is undertaken to compare groups, or at the request of the chief investigator, which is supported by survey responses indicating a preference for simple approaches from both groups. It could also be that the survey participants were restricted to those working in CTUs and industry and are perhaps not fully representative of those undertaking and reporting clinical trial results.

Many methods have been specifically proposed for adverse event analysis in RCTs, and there was a moderate level of awareness of these methods (38%), but in line with our review of journal articles we found uptake to be minimal (13%).[6 7] While not directly comparable, our results are also closely aligned with the results of a survey of industry statisticians and clinical safety scientists, undertaken by Colopy et al[28] that indicated a reliance on traditional methods for descriptive statistics and frequentist approaches when analysing harm outcomes.

This survey did not specifically ask participants about their use of graphics to display adverse event data, but a similar proportion of participants indicated the use of such summaries in free text comments as identified in our review of journal articles (9% vs 12%).[1] However, these figures were both substantially lower than the 37% that indicated the use of static visual displays for study level adverse event analysis in the survey of industry statisticians.[28] This could reflect the use of more advanced graphical approaches for internal reports and the widespread investment in data visualisation by industry as evidenced by the emergence of departments dedicated to data visualisations within many pharmaceuticals.

Education via training and guidance for statisticians and trialists about appropriate adverse event analysis could lead to improved practice and were both strongly endorsed as solutions by participants of both the survey and the workshop. Guidelines such as the harms extension to Consolidated Standards of Reporting Trials (CONSORT); the pharmaceutical industry standard from the Safety Planning, Evaluation and Reporting Team (SPERT); and the joint pharmaceutical/journal editor collaboration guidance on reporting of harm data in journal articles already exist and make several recommendations for analysing adverse events.[17 18 29] However, adherence to the CONSORT Harms checklist has been shown to be poor; and while the impact of the Lineberry et al[18] guidance and the Crowe et al[29] guidance has not been formally evaluated, our review of adverse event analysis practices indicates uptake of suggestions within these guidelines such as 'reporting CIs around absolute risk differences' and to 'include both the number of events (per person time) and the number of patients experiencing the event' to be minimal.[1 2 4 14 15] It has also been argued that such guidelines do not go far enough and fail to account for the complex nature of harm outcomes data.[5] Tutorial papers or case studies detailing examples of appropriate analysis could lead to wider adoption of such recommendations and to improvements in analysis practices, and development of such resources was highlighted as a priority by workshop participants. While the acquirement of the necessary knowledge and skills to implement new methods is essential, so too is increasing awareness of good practices and alternative methods. Guidance or tutorial papers can be useful to increase knowledge, but wide dissemination and promotion to encourage use are vital if we are to improve practice.

A change in attitude from both statisticians and the wider research community away from doing what they have always done is also needed. Journals and regulators

play a leading role in influencing good practice and could influence statisticians' and trialists' practice through policy change. *The New England Journal of Medicine* has already updated their policy to demand that evidence about both benefits and harms of treatments includes point estimates and margins of error and requires no adjustment for multiplicity where significance tests are performed for harm outcomes 'because information contained in the safety endpoints may signal problems within specific organ classes, the editors believe that the type I error rates larger than 0.05 are acceptable'.[30] A journal-wide initiative to adopt existing guidelines, for example, through the mandatory submission of the CONSORT Harms checklist would be one simple, initial step towards change.

Trial design and the nature of adverse event outcomes can also hinder the analyses performed. Unlike efficacy outcomes, which are well-defined and limited in number from the outset, harm outcomes are numerous, undefined and contain additional information on severity, timing and duration, and number of occurrences, which all need to be considered. More careful consideration of harm outcomes when designing, analysing and reporting trials will help produce a more balanced view of benefits and risks.

Improved analysis could be achieved through adoption of existing or development of more appropriate methods for adverse event data. Several participants mentioned adverse event analysis approaches we believe warrant exploring, including time-to-event analyses, data visualisations and Bayesian methods. Ultimately, with the aim of helping to identify signals for adverse drug reactions enabling a clearer harm profile to be presented. This is supported by feedback obtained at the workshop and the earlier findings of Colopy *et al*[28] who concluded that statisticians should help 'minimise the submission of uninformative and uninterpretable reports' and thus present more informative information regarding likely drug–event relationships.[28]

Participants of both the survey and the workshop raised concerns about the quality and reporting of adverse event data from RCTs. We agree that if adverse event data are not robust the analysis approach is redundant as the results will not be accurate. Therefore, procedures should be put in place at the trial design stage to mitigate problems with adverse event data collection, including, for example, development of validated methods for data collection and clear, standardised instructions for those involved in the detection and collection.[3 31]

### Strengths and limitations

Through support of the UKCRC CTU network and utilisation of personal contacts, we were able to achieve a high response rate for the survey. After invitations were sent, there was no way to ensure that responses were restricted to one per unit or organisation. However, dissemination via the UKCRC to senior statisticians within units and personal, senior contacts within industry would have

ensured some quality control. There was some level of self-selection for those recruited via the open platform, and as such, there is a possibility that these participants had an increased interest in adverse event analysis and are not fully representative of the clinical trial community. We also did not have any information on non-responders and as such cannot characterise any potentially relevant differences that could affect the generalisability of our results. This survey provides insight and essential starting points to identify areas of focus to help support a change to improve adverse event analysis practice. Many of the opinions raised in the survey were echoed by the workshop attendees who represented more of a general interest group.

### CONCLUSIONS

This research demonstrates that there is a moderate level of awareness of appropriate statistical methods for adverse event analysis but that these methods are not being used by statisticians and supports our earlier work identifying adverse event analysis practices in RCTs as suboptimal. Participants made a unanimous call for guidance on appropriate methods for adverse event analysis and training to support change. Feedback from both survey and workshop participants is that further research is needed to identify the most appropriate statistical methods for adverse event data analysis from all those available.

**Acknowledgements** The authors would like to thank: Dr Odile Sauzet for discussing the survey design in the initial stages of development. Louise Williams (UKCRC) for her support, in particular circulating the survey to UKCRC registered CTUs and helping us achieve such a high response rate, and Professor Carrol Gamble and Professor Catherine Hewitt (UKCRC statistical operational group chairs) for supporting this project. Alexander Schacht for inviting us on to his podcast to promote the survey and circulating to his contacts through LinkedIn. Dr Suzie Cro, Nicholas Johnson, Anca Chis Ster, Emily Day, Fiona Reid and Professor Carrol Gamble for providing feedback on survey content and platform. Also to Dr Suzie Cro for her help in facilitating the workshop at the UKCRC biannual statisticians' operations group meeting.

**Contributors** RP and VC conceived the idea, designed and ran the survey. RP performed the data analysis, interpreted the results and wrote the manuscript. VC interpreted the results, provided critical revision of the manuscript and supervised the project.

**Funding** This research was supported by the NIHR grant number DRF-2017-10-131.

**Disclaimer** This paper presents independent research funded by the National Institute for Health Research (NIHR). The views expressed are those of the author(s) and not necessarily those of the NHS, the NIHR or the Department of Health and Social Care.

**Competing interests** None declared.

**Patient and public involvement** Patients and/or the public were involved in the design, or conduct, or reporting, or dissemination plans of this research. Refer to the Methods section for further details.

**Patient consent for publication** Not required.

**Ethics approval** This study was granted ethical approval by the Imperial College Joint Research Compliance Office (ICREC reference: 19IC5067).

**Provenance and peer review** Not commissioned; externally peer reviewed.

**Data availability statement** Data are available in a public, open access repository https://doi.org/10.6084/m9.figshare.12436574.v1

**ORCID iD**
Rachel Phillips http://orcid.org/0000-0002-3634-7845

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
