## [Reviewer comments · BMJ Open]

ARTICLE DETAILS

TITLE (PROVISIONAL)	Understanding current practice, identifying barriers and exploring priorities for Adverse Event analysis in Randomised Controlled Trials: an online, cross-sectional survey of statisticians from academia and industry
AUTHORS	Phillips, Rachel; Cornelius, Victoria

VERSION 1 – REVIEW

REVIEWER	Paul Lee Hong Kong Polytechnic University, Hong Kong
REVIEW RETURNED	06-Feb-2020

GENERAL COMMENTS	General comments This survey collected information on statisticians on their practice and opinions on adverse event analysis in RCTs. Results of this survey were useful to increase the awareness in AE analysis, and to provide directions in the formation of AE analysis guidelines. I only have some minor concerns on the manuscript. Specific comments Page 3, line 45. What was the total sample size? Page 4, lines 18-20. I don't believe there exist a "most appropriate statistical method" for AE data analysis. Page 6, line 33-40. Please quantify how frequent these AE analysis were presented in RCT reports. Page 10, line 3. The opening time belonged to methods (please move to page 8, line 44). How did the participants consented to participate (did they consent by submitting the completed questionnaire or they had to complete a consent form)? Page 11, line 13. Frequencies and percentages are used to summarize the AE in one group, while risk difference, odds ratio, risk ratio, and incidence rate ratio are used to summarize the difference of AE between two groups. Table A1. What is meant by "Speciality"? Table A2. Please indicate that the options under "Summary statistic" and "AE comparison" allow multiple selections. Tables A3, A6 - A11. In some questions, the option "Don't know" was in the first row, while in some questions this option was in the last row. Please be consistent.
--

REVIEWER	Jennifer Thompson London School of Hygiene and Tropical Medicine, UK
REVIEW RETURNED	17-Apr-2020

GENERAL COMMENTS	This article reports the results of a survey of statisticians on their analysis of adverse events and barriers to improving this analysis. The research is useful to inform ways forward to improve practice
--

in this area, however the paper could be written much more succinctly and I have concerns that the conclusions do not follow from the research. Below are some more detailed comments

Please write out adverse event and randomised controlled trial in the title.

Abstract:

It isn't immediately clear what the percentages are in the abstract results (the 75%, 71%)

Introduction:

I think you have assumed too much knowledge of the reader. The introduction needs to include (1) what other reviews have found is done in practice (2) what the problems are with this (3) what the published improved methods are to over these problems (even if it's just to name the methods). I finished the paper without knowing what I should or shouldn't do to analyse AE.

Results:

The results could be written much more succinctly with less detail of very rare responses.

I was more interested in some of the appendix tables than table 1-3. Considering moving some appendix tables to the main paper and main tables to the appendix

The methods listed by your participants: it wasn't clear if these are good methods to use or not. Some of them seems like standard methods, like time to event analysis and incidence rates.

Discussion:

Your main conclusions are:

- "Further research is needed to identify the most appropriate statistical methods for AE data analysis" your research doesn't look at what methods are available and how appropriate they are, so this can't be a conclusion of your research.
- A call for more guidance. I disagree with this. It sounds like there is a lot of guidance already and the problem is that no one is aware of it

In my opinion, the main barriers are awareness of methods (only 40% were aware of methods), and issues with time and complexity for those that are aware. Tutorials and increasing knowledge of guidance might help with the first problem. The second is more complex.

	Paragraph 2 page 22: 32% seems pretty similar to 38% when you have a small sample. Reason is probably that your sample is not representative. Trials are not just conducted by trials units and industry Page 22 line 57: is this reliance on traditional methods for AEs? Optional comments  - The definition of AE and ADRs could go in the main text? - Table A1: what does specialty mean?
--	---

VERSION 1 – AUTHOR RESPONSE

Reviewer: 1

Reviewer Name: Paul Lee

Institution and Country: Hong Kong Polytechnic University, Hong Kong Please state any competing interests or state 'None declared': None declared

General comments

1. This survey collected information on statisticians on their practice and opinions on adverse event analysis in RCTs. Results of this survey were useful to increase the awareness in AE analysis, and to provide directions in the formation of AE analysis guidelines. I only have some minor concerns on the manuscript.

Response to reviewer:

We would like to thank the reviewer for their comments, which helped identify some ambiguities within our manuscript which we have now clarified in the resubmission. We address each of the points directly below.

Specific comments

2. Page 3, line 45. What was the total sample size?

Response to reviewer:

We thank the reviewer for bringing this omission to our attention. We have now added the total sample size into the results section of the abstract.

Revised text:

Abstract results (page 2): "Of the 64 participants that took part, forty-six participants were classified as public sector participants and eighteen as industry participants."

3. Page 4, lines 18-20. I don't believe there exist a "most appropriate statistical method" for AE data analysis.

Response to reviewer:

We agree with the reviewer that there is no one appropriate statistical method for AE data analysis. However, there a range of potential methods that take into consideration the complex nature of AE data, but further work is needed to identify which methods are the most appropriate. We have amended the text of the conclusion in the abstract to reflect this.

Revised text:

Abstract conclusion (page 3): “Improvements are needed and further research in this area is required to identify appropriate statistical methods. This research provides a unanimous call for the

development of guidance, as well as training on suitable methods for AE analysis to support change.”

4. Page 6, line 33-40. Please quantify how frequent these AE analysis were presented in RCT reports.

Response to reviewer:

A systematic review undertaken by the authors found no examples of signal detection methods being used for AE analysis. We found authors preferred to summarise binary outcomes using frequencies (94%) and percentages (87%), with only 9% presenting incidence rates. This is supported by a more recent review from Patson et al., we have added this extra information and reference to our article.

Revised text:

Introduction (page 5): “Previous research has demonstrated that these methods are not used for the analysis presented in the primary results publication. In a recent systematic review of 184 published reports in high impact journals there are no examples of these proposed methods being used, with authors preferring simple approaches predominantly presenting frequencies and percentages of events.^{1, 5} The statistical methods proposed for adverse event analysis identified in the methodology review had minimal citations, which further suggests uptake of these methods is low.^{1, 13, 14}”

5. Page 10, line 3. The opening time belonged to methods (please move to page 8, line 44). How did the participants consented to participate (did they consent by submitting the completed questionnaire or they had to complete a consent form)?

Response to review:

We thank the reviewer for pointing out this error with the time information and have moved the necessary text to the methods section and deleted it from the results section.

We also thank the reviewer for bringing our attention to the omission of the consent information. We have added a paragraph to the ‘sampling and recruitment’ section of the methods section clarifying this.

Revised text:

Methods - sampling and recruitment (pages 7-8): “The survey opened in April 2019 and remained open for 8 weeks. We also created an open platform for participants that was promoted at the June 2019 Promoting Statistical Insights (PSI) conference, the Effective Statistician podcast broadcast in July 2019, and Twitter and LinkedIn platforms. This platform remained open for 10 weeks.”

“The invitation to participate in the study included the participant information sheet, which was also included at the beginning of the survey before participants formally entered. Participants were encouraged to read the information sheet and discuss the study with others or contact the research team if they wished. If invitees were happy to enter into the trial at that point their consent was taken as implied upon submission of the completed survey.”

6. Page 11, line 13. Frequencies and percentages are used to summarize the AE in one group, while risk difference, odds ratio, risk ratio, and incidence rate ratio are used to summarize the difference of AE between two groups.

Response to reviewer:

We agree with the reviewer that these items are used to convey slightly different pieces of information. Participants were asked how they summarised AE data and were able to select multiple responses from this list that contain both descriptive and summary pieces of information hence they are reported together i.e. a responder might present frequencies, percentages and risk-difference. As such, we have amended the description in table A2 to indicate the information displayed is both descriptive and summative. As requested by reviewer 2 this table now appears as table 2 in the revised manuscript.

Revised text:

Results Table 2 (page 12): “Descriptive and summary statistics[†]”

7. Table A1. What is meant by “Speciality”?

Response to reviewer:

We would like to thank the reviewer for highlighting this ambiguity. Participants were asked if there was a clinical area they predominantly worked on and we use the term “speciality” to refer to this. We have added text to the footnote of table A1 to clarify this. As requested by reviewer 2 this table now appears as table 1 in the revised manuscript.

Revised text:

Results Table 1 (page 10): “[†]Participants were asked if there was a clinical area they predominantly worked on.”

8. Table A2. Please indicate that the options under “Summary statistic” and “AE comparison” allow multiple selections.

Response to reviewer:

We have added this clarification to the footnote of table A2. As requested by reviewer 2 this table now appears as table 2 in the revised manuscript.

Revised text:

Results Table 2 (page 12): “[†] Participants were able to provide multiple responses to this question”

9. Tables A3, A6 - A11. In some questions, the option “Don’t know” was in the first row, while in some questions this option was in the last row. Please be consistent.

Response to review:

We thank the reviewer for drawing our attention to this inconsistency and have amended it so that the “don’t know” category is always displayed in the last row.

Reviewer: 2

This article reports the results of a survey of statisticians on their analysis of adverse events and barriers to improving this analysis. The research is useful to inform ways forward to improve practice in this area, however the paper could be written much more succinctly and I have concerns that the conclusions do not follow from the research. Below are some more detailed comments

Response to reviewer:

We would like to thank the reviewer for their helpful comments which we address directly below.

1. Please write out adverse event and randomised controlled trial in the title.

Response to reviewer

We have removed acronyms from the study title and amended it to include the study design i.e. that it was an online, cross-sectional survey as also requested by the editor.

Revised text:

Title: "Understanding current practice, identifying barriers and exploring priorities for Adverse Event analysis in Randomised Controlled Trials: an online, cross-sectional survey of statisticians from academia and industry"

Abstract:

2. It isn't immediately clear what the percentages are in the abstract results (the 75%, 71%)

Response to reviewer:

We thank the reviewer for drawing our attention to this ambiguity and have amended the text to clarify this statement in both the abstract and results section.

Revised text:

Abstract results (2): "Thirty-eight (38/51; 75%) CTUs, five (5/7; 71%) industry and twenty-one attendees at the 2019 PSI conference participated in the survey."

Results (page 9): "Invitations were sent to fifty-one CTU/public sector and seven industry contacts. Thirty-eight (75%) units and five (71%) industry contacts participated in the survey giving an overall response rate of 74%. Twenty-four people consented to participate via the open platform, of which 21 participated in the survey."

Introduction:

3. I think you have assumed too much knowledge of the reader. The introduction needs to include (1) what other reviews have found is done in practice (2) what the problems are with this (3) what the published improved methods are to overcome these problems (even if it's just to name the methods). I finished the paper without knowing what I should or shouldn't do to analyse AE.

Response to reviewer:

We thank the reviewer for drawing our attention to this and pointing out what further information would be helpful. We have rewritten and expanded the introduction, agreeing with the reviewer that we assumed too much prior knowledge in this first iteration. We believe the edited version now gives a more thorough background.

Revised text:

We have not included the entire rewrite here for brevity but changes are tracked within the resubmission and cover pages 5-6.

Results:

4. The results could be written much more succinctly with less detail of very rare responses.

Response to reviewer:

We thank the reviewer for highlighting this. We have removed several comments from the results section, which we hope addresses the reviewer's concerns whilst still providing a sufficient summary of our results. We feel that some of the comments regarding the rare responses are useful as they offer unique insight into practices and statisticians' thinking so we have retained some of this information but in a more concise manner.

Revised text:

We have not included rewrite her for brevity but changes are tracked within the resubmission and cover pages 9-16.

5. I was more interested in some of the appendix tables than table 1-3. Considering moving some appendix tables to the main paper and main tables to the appendix

Response to reviewer:

We thank the reviewer for this insight. We have moved tables 1-3 from the main article to the appendix and tables A1-A3 into the main article, combining tables A2 and A3 into table 2. We have retained figures 1 and 2 in the main article.

6. The methods listed by your participants: it wasn't clear if these are good methods to use or not. Some of them seems like standard methods, like time to event analysis and incidence rates.

Response to reviewer:

Some of the methods mentioned by participants are standard statistical approaches used for efficacy analysis but are rarely used for AEs. We try to point out in the discussion which of the methods mentioned by participants we advocate and state "Several participants mentioned AE analysis approaches we believe warrant exploring including time-to-event analyses, data-visualisations and Bayesian methods." We have also revised the text in the results section to try to make it clearer which methods we believe are suitable.

Revised text:

Results (page 12-13): "Just under 40% stated that they were aware of appropriate methods published specifically for AE analysis in RCTs (appendix table 2)."

Discussion:

7. Your main conclusions are:

- "Further research is needed to identify the most appropriate statistical methods for AE data analysis" your research doesn't look at what methods are available and how appropriate they are, so this can't be a conclusion of your research.
- A call for more guidance. I disagree with this. It sounds like there is a lot of guidance already and the problem is that no one is aware of it

Response to reviewer

The reviewer is correct to point out that our first conclusion about further research being needed did not come directly from the survey results. However, we did receive this feedback from workshop participants, which we had not fully summarised in the results section. We have now added this additional material to make it clearer.

Our second conclusion regarding the need for guidance is directly taken from survey responses (100% of participants called for guidance) and was supported by workshop participants. Whilst it is true that some guidance already exists, it is limited in terms of recommendations for analysis methods e.g. CONSORT harm focuses on how trialists should report analysis not what analysis should be undertaken so we believe our conclusion is appropriate. In addition, a recent paper by Patson et al. has also argued that statistical methods used “rarely account for the complexity of the collected safety data”.

We agree with the reviewer that lack of awareness of any guidance, existing or new is a problem and we acknowledge this in the discussion “Whilst the acquirement of the necessary knowledge and skills to implement new methods is essential, so too is increasing awareness of good practices and alternative methods. Guidance or tutorial papers can be useful to increase knowledge, but wide dissemination and promotion to encourage use is vital if we are to improve practice.”

Revised text

Abstract – Conclusion (page 3): “Improvements are needed and further research in this area is required to identify appropriate statistical methods. This research provides a unanimous call for the development of guidance, as well as training on suitable methods for AE analysis to support change.”

Results (page 16): “In discussions, the following themes were highlighted as priorities to take forward: development of guidelines; identification of appropriate analysis methods; exploring integration of qualitative information; and ensuring consistency of information reported including development of core harm outcomes by drug class.”

Discussion (page 18): “It has also been argued that such guidelines do not go far enough and fail to account for the complex nature of harm outcomes data.⁵”

Reference: “5. Patson N, Mukaka M, Otwombe KN, et al. Systematic review of statistical methods for safety data in malaria chemoprevention in pregnancy trials. *Malaria Journal* 2020; 19: 119. DOI: 10.1186/s12936-020-03190-z.”

Conclusion (page 21): “This research demonstrates that there is a moderate level of awareness of appropriate statistical methods for adverse event analysis but that these methods are not being used by statisticians and supports our earlier work identifying adverse event analysis practices in RCTs as sub-optimal. Participants made a unanimous call for guidance on appropriate methods for adverse event analysis and training to support change. Feedback from both survey and workshop participants is that further research is needed to identify the most appropriate statistical methods for adverse event data analysis from all those available.”

8. In my opinion, the main barriers are awareness of methods (only 40% were aware of methods), and issues with time and complexity for those that are aware. Tutorials and increasing knowledge of guidance might help with the first problem. The second is more complex.

Response to reviewer:

We agree with the reviewer that these are some of the barriers and we state in the discussion that “the acquirement of the necessary knowledge and skills to implement new methods is essential, so too is increasing awareness of good practices and alternative methods. Guidance or tutorial papers can be useful to increase knowledge, but wide dissemination and promotion to encourage use is vital if we are to improve practice.” However, we did not make it clear in our first submission that feedback particularly from workshop participants is that we don’t know how good these methods are and that further research is needed to assess which methods to use and when. We have now amended the text to include these results.

Revised text

Results (page 16): “In discussions, the following themes were highlighted as priorities to take forward: development of guidelines; identification of appropriate analysis methods; exploring integration of qualitative information; and ensuring consistency of information reported including development of core harm outcomes by drug class.”

9. Paragraph 2 page 22: 32% seems pretty similar to 38% when you have a small sample.
Reason is probably that your sample is not representative. Trials are not just conducted by trials units and industry

Response to reviewer:

38% is the lower end of the range of values identified in reviews. The upper range is 47%, so whilst not vastly different the reported use in this survey is lower than the range of values reported across the board by review articles. However, we agree that journal articles represent the entire range of trials that won’t necessarily be performed by units or industry so we have added this as a potential reason into the text.

Revised text:

Discussion (page 17): “Reported use of 95% confidence intervals were similar to that reported in journal articles (22% compared to 20%) but reported use of hypothesis testing was lower than what was found in journal articles (32% compared to a range of 38% to 47%).¹⁻³ Reasons for these disparities are not known but could include journals editors requesting such analyses is undertaken to compare groups, or at the request of the chief investigator, which is supported by survey responses indicating a preference for simple approaches from both groups. It could also be that the survey participants were restricted to those working in CTUs and industry, and are perhaps not fully representative of those undertaking and reporting clinical trial results.”

10. Page 22 line 57: is this reliance on traditional methods for AEs?

Response to reviewer:

Thank you for highlighting this ambiguity. This paper was referring to the analysis of harm outcomes and we have added the following text to clarify this.

Revised text:

Discussion (page 17-18): “Whilst not directly comparable, our results are also closely aligned with the results of a survey of industry statisticians and clinical safety scientists, undertaken by Colopy and colleagues that indicated a reliance on traditional methods for descriptive statistics and frequentist approaches when analysing harm outcomes.²⁵”

Optional comments

11. The definition of AE and ADRs could go in the main text?

Response to reviewer:

We agree with the reviewer and have amended the text accordingly.

Revised text:

Introduction (page 5): “Adverse events are events that may or may not be related to the treatment under investigation, and adverse drug reactions are events classified as related to the treatment under investigation.”

12. Table A1: what does specialty mean?

Response to reviewer:

We would like to thank the reviewer for highlighting this ambiguity. Participants were asked if there was a clinical area they predominantly worked on and we use the term “speciality” to refer to this. We have added text to the footnote of table A1 to clarify this. As per earlier comments this table now appears as table 1 in the main text.

Revised text:

Table 1 (page 10): “¹Participants were asked if there was a clinical area they predominantly worked on.”

VERSION 2 – REVIEW

REVIEWER	Paul Lee Hong Kong Polytechnic University, Hong Kong
REVIEW RETURNED	12-May-2020
GENERAL COMMENTS	All my comments had been addressed in this revision.
REVIEWER	Dr Jennifer Thompson London School of Hygiene and Tropical Medicine
REVIEW RETURNED	15-May-2020
GENERAL COMMENTS	I would like to thank the authors for the significant amendments to their article. They have addressed all of my comments and I recommend it is accepted for publication.